# Auditing the readiness of healthcare facilities for referral and management of pre-eclampsia cases in Zanzibar- a study protocol

**Salma A. Rashid** [1]*, **Saada A. Seif** [2]

1 Department of Clinical Nursing, The University of Dodoma, Dodoma, Tanzania, 2 Department of Nursing Management and Education, The University of Dodoma, Dodoma, Tanzania

☯ These authors contributed equally to this work.

\* sallyrasheed.sr@gmail.com

## Abstract

Hypertensive disorders of pregnancy are reported as the second leading root of maternal morbidity and mortality in Zanzibar. Evidence shows that the majority of pregnant women in Zanzibar are referred late from lower-level healthcare facilities, and majority develop complications of eclampsia. This study's goal is to determine if all public healthcare facilities in Zanzibar are prepared to manage pre-eclampsia cases and if lower-levels public healthcare facilities are ready to refer pre-eclampsia cases. This will be a descriptive cross-sectional study that will involve a total of 54 healthcare facilities and 176 health care providers working in antenatal clinics. All public health care facilities will be stratified into tertiary, secondary, and primary strata. A simple random sampling will be used to select 46 healthcare facilities in the primary stratum while all healthcare facilities within the tertiary and secondary strata will be selected. In each healthcare facility, a physical observation will be performed to assess the availability of equipment and supplies, medications, and lab tests, while a self-administered questionnaire will be used to assess the knowledge level and skills of healthcare providers for the management of pre-eclampsia and eclampsia. Patient's case files in the tertiary and secondary strata will be reviewed to assess the quality of management of pre-eclampsia while the service records of the primary stratum will be assessed for compliance status with referral guidelines. Data will be analyzed using SPSS version 25. Descriptive statistics will be used to describe the frequency distribution of the study variables, and results will be presented in terms of frequency and percentage. The Chi-square test will be used to describe the relationship between variables, and a p-value of < 0.05 will be regarded as a statistically significant difference.

## Introduction

Pre-eclampsia is defined as a pregnancy complication characterized by high blood pressure of at least 140 mmHg systolic and 90 mmHg diastolic pressure measured on two occasions, 4–6 hours apart, accompanied by proteinuria of at least 300 mg per 24 hours, or at least +1 on dipstick tested after 20 weeks of gestation age. In severe cases, it may be accompanied by swelling

**Data Availability Statement:** No datasets were generated or analysed during the current study. All relevant data from this study will be made available upon study completion.

**Funding:** The authors received no specific funding for this work.

**Competing interests:** The authors have declared that no competing interests exist.

of lower limbs, severe headache, upper abdominal pain, blurred vision or light sensitivity. In women with pre-eclampsia, eclampsia is defined as the presence of a new-onset grand mal seizure [1]. The pathogenesis of pre-eclampsia is unknown, but it is associated with abnormal placentation at the first half of pregnancy, followed by hypertension and proteinuria in the second half [2]. Risk factors like obesity, chronic hypertension, diabetes, nulliparity, and twinning pregnancies have been shown to increase the likelihood of pregnant women developing pre-eclampsia [3].

The prevalence of pre-eclampsia and eclampsia is reported to be much higher in developing countries, where it ranges from 1.8 to 16.7% [4], while in developed countries like United States of America, the pre-eclampsia prevalence ranges from 3–6% [5]. In East African countries, the prevalence of pre-eclampsia ranges from 6.1% in Kenya [6] to 12.2% in Ethiopia [7]. In Tanzania the prevalence of severe pre-eclampsia was found to be 4.2% [8], while in Zanzibar, the prevalence of pre-eclampsia was reported to be 9% in the year 2011 [9].

Research shows that pre-eclampsia and eclampsia are among the primary causes of maternal and neonatal morbidity and mortality. It is reported to affect 2–8% of pregnancies globally, and it is associated with 10–15% of direct maternal deaths and up to 25% of stillbirths and newborn deaths in developing countries [10, 11]. Tanzania is reported to have 556 maternal deaths per 100,000 live birth [12], and eclampsia related complications account for 18.9% of maternal deaths [13]. In Zanzibar, the 2016 report showed that maternal mortality was 287 deaths per 100,000 live births [14], while the 2017 report showed that severe pre-eclampsia was found to contribute to 25.8% of maternal morbidity and 21.8% of maternal mortality [15].

The maternal deaths due to eclampsia could be prevented if pregnant women could access quality antenatal care (ANC) services [16]. The quality of ANC services depend on the availability of competent healthcare providers and well-functioning healthcare facilities [17] [18]. Unavailability of equipment, medication and supply and lack of healthcare providers with adequate knowledge and skills for the management of pre-eclampsia may result in failure to early detect, diagnose and provide appropriate treatment and or referral of pre-eclampsia cases [19, 20]. Therefore, assessing their availability in every healthcare facility level is very crucial.

All pregnant women attending ANC must be screened for pre-eclampsia or eclampsia, and those showing signs of pre-eclampsia or eclampsia, must be appropriately managed. Those presenting with severe features of pre-eclampsia at lower healthcare facility levels must be referred to the next level immediately using the referral guideline, while the most complicated cases have to be referred to a tertiary hospital for further management [19]. Despite these clear established pathways, evidence showed that the majority of pregnant women with severe features of pre-eclampsia in Zanzibar are referred late to the higher levels, and the majority develop complications of eclampsia (25.8%) [15].

This has made us to question whether all levels of healthcare facilities in Zanzibar are adequately prepared in terms of having the tools and medications on hand, adhering to management and referral guidelines, as well as having healthcare providers with knowledge and skills for the management of pre-eclampsia/eclampsia. Therefore, this study intends to evaluate the healthcare facilities' preparedness for ANC referral and management of pre-eclampsia cases in Zanzibar.

## Theoretical model: Donabedian model for measuring quality of care

The Donabedian model for measuring quality care (Fig 1) will guide this study [20]. This model has three approaches to evaluating the quality of care: structure, process, and outcome. The structure which is also known as input measure, entails the characteristics of the healthcare facility, such as staff-to-patient ratio and operating the times and hours of the services.

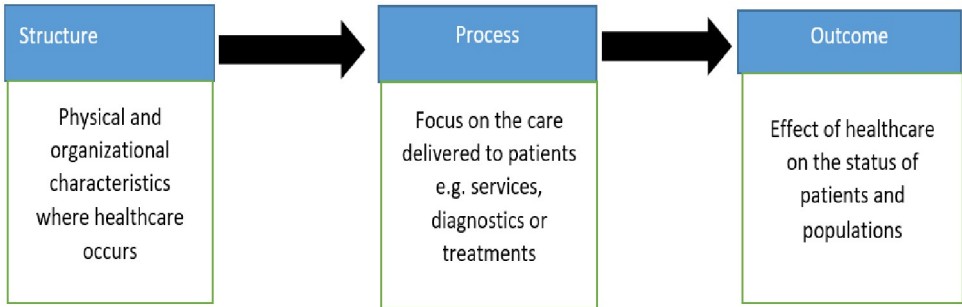

**Fig 1. This a Donabedian model for measuring the quality care.**

Process measures reflect the way systems work to deliver the desired outcome, and the outcome measure reflects the impact on the patient and demonstrates the result of improvement and whether it has ultimately achieved the aim(s) set [20].

The current study will adopt the structure and the process domains only. The structure measures will be the availability of resources in the health care facilities, like skilled and knowledgeable health care providers, availability of guidelines, availability of required tests, resources and equipment, on-the-job training, availability of ambulance, and availability of medication. The process measure will be adherence to the guideline for the treatment of pre-eclampsia and eclampsia, as well as adherence to the guidelines for referring women with signs of severe pre-eclampsia.

## Methods

### Study aim

The purpose of this study is to assess the availability of pre-eclampsia management equipment, medication, tests, and guidelines, as well as health care providers working in ANC with adequate knowledge and skills in pre-eclampsia management at all healthcare facility levels. In addition, this study will assess the quality of pre-eclampsia and eclampsia management at higher healthcare facility levels and the compliance of referral guidelines at lower healthcare facility levels.

### Study design

A descriptive cross-sectional design using a quantitative approach will be used.

### Study area

This study will be conducted in Zanzibar, which is united with and a part of Tanzania. The study setting will be healthcare facilities of all levels in Zanzibar. The healthcare delivery system in Zanzibar is organized into three levels, which are the primary level, secondary and tertiary level. The primary level is the lowest level, which includes Primary Health Care Unit (PHCU), Primary Health Care Unit Plus (PHCU +), and Primary Health Care Center or Cottage (PHCC). In total, there are 122 PHCU, 34 PHCU+, and 4 PHCC. Primary health care units provide primary health care services and all ANC services, including screening of pre-eclampsia, while the PHCU+ provides additional services such as delivery, dental, laboratory, and pharmaceutical services while the PHCCs provide the same services as PHCU+ with the addition of inpatient and X-ray services.

The secondary level, which is the middle level of health care delivery, includes district hospitals and the regional hospitals. There are two district hospitals and one regional hospital. District hospitals provide second-line referral services, including basic surgery, and regional hospitals provide referral points for all district hospitals. The highest level of health care delivery is the tertiary level, also called the referral hospital. There is only one referral hospital in Zanzibar, which is Mnazi Mmoja National Hospital, which provides referral services from the district and regional hospitals.

## Study population

This study will involve two types of study populations: all healthcare facilities at the primary, secondary and tertiary levels, and healthcare providers working in ANC units at all healthcare facility levels.

## Sample size estimation

**Sample size estimation for the health care providers.** The sample size of the health care providers was calculated by using the Cochran formula [21]. The proportion of health care providers with knowledge on the management of pre-eclampsia of 11.8% [22], a Z-score value of 1.96 and a margin error of 5% was used. Adding 10% of the none response, the minimum sample size of 176 healthcare providers will be used.

**Sample size estimation of healthcare facilities.** The sample size will be calculated using a formula put forward by WHO, which states that in assessing lower health facility levels, if there are 100 or fewer, you should study all of them. If there are more than 100, then take at least 30% [23]. In this regard, the total number of primary healthcare units in Zanzibar (PHCUs and PHCU+) is 154: therefore, only 30% will be taken, which is equal to 46 primary healthcare units. Moreover, the total number of primary health care centers is 4, the district hospital is 2, the regional hospital is 1, and the referral hospital is 1, so all of them will be included in the study. Therefore, the total number of healthcare facilities that will be included in this study will be 54.

## Sampling procedure

All public health care facilities will be stratified into tertiary, secondary, and primary strata. The healthcare facilities within the tertiary and secondary strata will all be studied, while a simple random sampling will be used to select 46 healthcare facilities in the primary stratum. Within each stratum, all healthcare providers working in ANC units will be conveniently selected.

## Measurement of Variables

**Quality of management of pre-eclampsia.** This refers to all treatment, measurements and tests performed on a pregnant woman with signs of pre-eclampsia in accordance with the MOHWZ guidelines. This variable will be measured by 16 YES/NO items on a binary scale assessing whether management of pre-eclampsia was conducted as per the established guidelines, which includes provision of correct medications, monitoring of blood pressure and fetal heart rate, checking protein in urine and conducting liver and kidney function tests [24]. Five items will be for mild pre-eclampsia and eleven items for severe pre-eclampsia. One point will be awarded for each correct act performed and 0 points for each incorrect or missed acts. For mild pre-eclampsia, a total score of 5 points will be regarded as proper management, and for severe pre-eclampsia, a total score of 11 points will be regarded as proper management, less than that is improper management.

**Compliance of referral guideline.**   When a healthcare facility meets all required referral standards to refer a woman with signs of pre-eclampsia or eclampsia based on the recommended guidelines it is considered a successful referral in this study. This will be measured by 8 YES/NO items in nominal scale assessing whether the referral requirements are met as per the established guideline. The tool will be adopted from WHO, (2013), and it includes assessing if the facility has an established clear referral pathway, provides pre-referral treatment to the patient, refers the patient accompanied by a healthcare provider, has referral guideline and has an ambulance.

**Knowledge of pre-eclampsia.**   This will be measured by 10 multiple-choice questions on a ratio scale. The standardized structured questionnaire will be adopted from Olaoye et al. [25], which has a Cronbach alpha was 0.72. One point will be awarded for the correct answer and zero points for the incorrect answer. An example of the question is 'What is the loading dose of magnesium sulfate?' The total score will be 10, and the mean score will be used to differentiate between those with adequate knowledge and inadequate knowledge. The same questionnaire will used to all categories of healthcare providers and the knowledge level will be presented for each category.

**Skills for management of pre-eclampsia.**   This will be measured by seven items in the form of task questions. The tool will be adopted from Jhpiego, 2017 [26]. Participants will be required to describe in detail all the important steps in carrying out a certain procedure related to the management of pre-eclampsia. An example of the question is 'What are the steps for providing the loading dose of magnesium sulphate?' One point will be awarded for the completeness of the mentioned steps and zero points for an incomplete, incorrect or missed step. The total score will be 7 and the mean score will be used to categorize those with adequate skills against those with inadequate skills.

**Availability of equipment, tests and medication.**   These will be measured by 27 items on dichotomous nominal scale on the availability of well-functioning equipment necessary for the management of pre-eclampsia (14 items, e.g. a working blood pressure machine, a working fetoscope etc), the availability of essential medications (10 items, e.g. methyldopa and diazepam), and the availability of tests needed for the management of pre-eclampsia (3 items, eg. protein in urine test and kidney function test). The tool will be adopted from Maembe and Pembe, 2015 [27].

**Background characteristics.**   The background characteristics of healthcare providers will be measured by variables from the literature, which will include years of working experience in ANC, the level of working healthcare facility, and whether they have attended training on the management of pre-eclampsia.

## Data collection procedure

Data will be collected within the selected healthcare facilities for the period of four weeks. The principal investigator and six research assistants who are registered nurses will collect the data. In this study, several methods of data collection methods will be used depending on the type of data needed to answer a particular study objective as follows:

**Questionnaire.**   A self-administered questionnaire will be used to obtain data from healthcare providers for assessing their knowledge and skills on the management of pre-eclampsia. Each study participant in each healthcare facility will be given a printed questionnaire, which will be collected after 30 minutes.

**Documentary review.**   This method will be used to obtain information on the management of pre-eclampsia, whereby the principle investigator or research assistant will review the case files of patients diagnosed with pre-eclampsia by the medical doctor, whether mild or severe, to look for management provided against the standard treatment guidelines.

**Observation method.** This method will be used to assessing the availability of equipment, tests, and medications related to the management of pre-eclampsia using a pre-prepared checklist. The same method will be used to assess whether the facility meets the referral standards. In this method, the principal investigator or research assistant will inspect the facility and record the available equipment, medication, tests, guidelines, and referral practices.

## Data management plan

Data obtained from participants will be stored in a secured computer with password that only the researcher's team will be able to access, and all University of Dodoma safety protocols will be followed in order to secure participants information.

## Data analysis plan

The Data will be analyzed using SPSS version 25. The data will be cleaned and checked for completeness before being subjected to models of analysis. Descriptive statistics will be used, and data will be reported for each healthcare facility level using frequency and percentage. The frequency of distribution of equipment, medications, and tests will be reported. The background characteristics of healthcare providers will be reported, and the frequency distribution of those who have attended the on-the-job training, those with adequate knowledge and skills, and their professional qualifications in each level of healthcare facility will be reported. Furthermore, the frequency distribution of referral requirement items in each primary healthcare facility will be reported. The frequency distribution of the type of management provided for the mild and severe pre-eclampsia reviewed cases will be reported and management will be compared between healthcare facilities of secondary and tertiary levels using a chi-square test. The level of significance will be sat at $p < 0.05$. To avoid hospital bias, the data collector will not be one of the nurses working in the chosen facility: respondents' confidentiality will be assured: and data will be analyzed by considering the level of healthcare facility, with findings presented separately for each level of healthcare facility. The limitation of this study is on the skills measurement; this study relied on subjective assessment rather than objective assessment, which may lead to overestimation of skills scores. If resources allow, one should consider the observation of actual practice to assess the skills, either in real clinical settings or in a simulation environment.

## Ethical consideration and declaration

Ethical permission for this study was obtained from the University of Dodoma Research Ethics Committee (UDOM-REC) with reference number MA.84/261/02/218 and Zanzibar health research institute (ZAHRI) with a reference number ZAHREC/04/ST/JAN/2021/02. A written consent will be given to health care providers after explaining the purpose of the study and being told that their participation is voluntary and they can withdrawal from the study at any.

## Status and timeline of the study

The study is expected to be conducted for a period of four weeks, and on the day of submission, the status was in planning for data collection.

**Strength and limitation.** The strength of this study is that it will involve all healthcare facilities in Zanzibar, from the lowest level to the highest level. This is important because the results that will be obtained will describe the true picture of the current status of pre-eclampsia management in healthcare facilities in Zanzibar. To the best of the authors' knowledge, this will be the first study in Zanzibar to audit healthcare facilities for pre-eclampsia management

and referral. This information will help the Ministry of Health-Zanzibar make proper decisions to strengthen healthcare services to pregnant women in terms of increasing the number of skilled health care providers, medications, equipment and tests. All these will contribute to minimizing the delay in receiving appropriate management, thereby decreasing the burden of pre-eclampsia and eclampsia complications.

## Dissemination plan

Findings obtained from this study will be disseminated to the University of Dodoma, the Ministry of health Zanzibar, and will be submitted for publication in peer-reviewed journal.

## How amendments to the study, including termination, will be dealt with

If any changes are made to the study, including the study's termination, this will be communicated to the journal auditoria, which will submit the changes or provide a reason for the termination.

## Supporting information

**S1 File. This is a questionnaire for assessing knowledge and skills of healthcare providers on the management of pre-eclampsia and eclampsia.**
(DOCX)

**S2 File. This is a checklist for assessing healthcare facilities for availability of medication, equipment and test.**
(DOCX)

**S3 File. This is a checklist for assessing management of pre-eclampsia.**
(DOCX)

**S4 File. This is a checklist for assessing the referral standards.**
(DOCX)

## Acknowledgments

We acknowledge the support of Tasakhtaa Global Hospital-Zanzibar for the financial support to undertake master degree which lead to the production of this article.

## Author Contributions

**Conceptualization:** Salma A. Rashid.

**Methodology:** Salma A. Rashid.

**Supervision:** Saada A. Seif.

**Writing – original draft:** Salma A. Rashid.

**Writing – review & editing:** Saada A. Seif.

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
