## [Decision Letter · Decision Letter 0]

11 Sep 2021

PONE-D-21-13246

Auditing the Antenatal Care Referral and Management of Pre-Eclampsia among Health Facilities in Zanzibar- A Protocol for Cross Sectional Study

PLOS ONE

Dear Dr. Rashid,

Thank you for submitting your manuscript to PLOS ONE. After careful consideration, we feel that it has merit but does not fully meet PLOS ONE’s publication criteria as it currently stands. Therefore, we invite you to submit a revised version of the manuscript that addresses the points raised during the review process.

We look forward to receiving your revised manuscript.

Kind regards,

Frank T. Spradley

Academic Editor

PLOS ONE

Journal Requirements:

2. Please include additional information regarding the survey or questionnaire to be used in the study and ensure that you have provided sufficient details that others could replicate the analyses. For instance, if you developed a questionnaire as part of this study and it is not under a copyright more restrictive than CC-BY, please include a copy, in both the original language and English, as Supporting Information. If the original language is written in non-Latin characters, for example Amharic, Chinese, or Korean, please use a file format that ensures these characters are visible.

3. Please state how you will validate the questionnaire prior to testing on study participants. Please provide details regarding the validation group within the methods section.

Reviewers' comments:

Reviewer's Responses to Questions

**Comments to the Author**

1. Does the manuscript provide a valid rationale for the proposed study, with clearly identified and justified research questions?

Reviewer #1: Yes

2. Is the protocol technically sound and planned in a manner that will lead to a meaningful outcome and allow testing the stated hypotheses?

Reviewer #1: Partly

3. Is the methodology feasible and described in sufficient detail to allow the work to be replicable?

Reviewer #1: Yes

4. Have the authors described where all data underlying the findings will be made available when the study is complete?

Reviewer #1: Yes

5. Is the manuscript presented in an intelligible fashion and written in standard English?

Reviewer #1: No

6. Review Comments to the Author

You may also provide optional suggestions and comments to authors that they might find helpful in planning their study.

Reviewer #1: Greetings,

I commend the authors for selecting a very important research area dealing with maternal and child health in a developing country.

This protocol has many dark areas which need to be addressed:

1. The document is full of English language grammatical errors, which should be corrected.

2. Scientific English language need to be improved throughout the paper.

3. Please use uniform method of references in paper.

4. Please elaborate the method sections for following points.

• Please describe who will collect data.

• Please add duration of study.

• Rewrite the dependent and independent variables, presently this part is written haphazardly.

• Please state the guidelines/definitions explicitly those are being used at your peripheral hospital while managing hypertensive disorder in pregnant women. Also state whether these guidelines are uniformly followed at all centers or there are some differences in the protocols.

• Please state whether some training/orientation program exist for the peripheral health workers for providing antenatal care.

• Please describe the process to avoid hospital bias.

• Please describe groups for statistical analysis and how the comparison will be performed.

Best wishes,

7. PLOS authors have the option to publish the peer review history of their article (what does this mean?). If published, this will include your full peer review and any attached files.

Reviewer #1: No

---

## [Author Response · Author response to Decision Letter 0]

4 Jan 2022

RESPONSE TO REVIEWER’S COMMENTS

We would like to take this opportunity to thank you all, editors and reviewers from PlosOne’ publication team for your efforts devoted to review our protocol. We appreciate for all the comments given. We have thoroughly responded to all reviewers’ comments. The table below is a summary of point by point responses for the reviewers.

COMMENTS RESPONSE

1. The document is full of English language grammatical errors, which should be corrected.

 The grammatical errors have been corrected

2. Scientific English language need to be improved throughout the paper.

 The scientific English language has been improved

3. Please use uniform method of references in paper.

 The Vancouver style is now used for the whole document

4. Please elaborate the method sections for following points.

• Please add duration of study.

 The duration of the study is added on page 12 line 317

• Please describe who will collect data.

 The data collection procedure is now improved to include those who will collect the data on page 12 line 318

• Rewrite the dependent and independent variables, presently this part is written haphazardly.

 This part has been re-written and made clear (page 10-11, line 271-315

• Please state the guidelines/definitions explicitly those are being used at your peripheral hospital while managing hypertensive disorder in pregnant women. Also, state whether these guidelines are uniformly followed at all centers or there are some differences in the protocols.

 The Ministry of Health Zanzibar developed a ‘Management Protocol for Emergency Obstetric and Newborn Care’ to be followed uniformly in all health centers of lower and high levels. This information is now included in page 5, line 142-144.

• Please state whether some training/orientation program exist for the peripheral health workers for providing antenatal care.

 Yes there was a training program on ‘Focused Antenatal Care’ that is provided to all health care providers working in ANC. This information is now included in page 5 line 144-145

• Please describe the process to avoid hospital bias.

 In order to avoid hospital bias, the following mechanism will be used:

• The data collector will not be among nurses working in the selected facility

• Confidentiality from respondents will be assured

• Data will be analyzed by considering the level of healthcare facility. That is the findings for each healthcare facility level will be analyzed separately (page 13, line 361-367)

• Please describe groups for statistical analysis and how the comparison will be performed.

 • Descriptive statistics will be used to describe the proportion of pre-eclampsia, the frequency distribution of equipment, medication, tests, referral standards, and knowledge and skills of health care providers on the management of pre-eclampsia.

• Chi-square test will be used to show association between:

o Pre-eclampsia vs. demographic characteristics of pregnant women

o health care facility levels vs. management of pre-eclampsia

o health care facility levels vs. availability of equipment, test and medication

o health care facility levels vs. availability of skilled and knowledgeable health care providers

o knowledge vs demographic characteristics of health care providers

o skills vs. demographic characteristics of health care providers 

(page 13, line 353-361)

---

## [Decision Letter · Decision Letter 1]

11 May 2022

PONE-D-21-13246R1Auditing the Readiness of Healthcare Facilities in Antenatal Care Referral and Management of Pre-Eclampsia in Zanzibar- A Protocol for Cross Sectional StudyPLOS ONE

Dear Dr. Rashid,

Thank you for submitting your manuscript to PLOS ONE. After careful consideration, we feel that it has merit but does not fully meet PLOS ONE’s publication criteria as it currently stands. Therefore, we invite you to submit a revised version of the manuscript that addresses the points raised during the review process.

The manuscript has been evaluated by one reviewer, and their comments are available below.

Whilst the manuscript is improved, the reviewers have raised a number of concerns that still need attention. 

Could you please revise the manuscript to carefully address the concerns raised?

We look forward to receiving your revised manuscript.

Kind regards,

Sebastian Shepherd

Staff Editor

PLOS ONE

Reviewers' comments:

Reviewer's Responses to Questions

**Comments to the Author**

1. Does the manuscript provide a valid rationale for the proposed study, with clearly identified and justified research questions?

Reviewer #1: Yes

2. Is the protocol technically sound and planned in a manner that will lead to a meaningful outcome and allow testing the stated hypotheses?

Reviewer #1: Partly

3. Is the methodology feasible and described in sufficient detail to allow the work to be replicable?

Reviewer #1: Yes

4. Have the authors described where all data underlying the findings will be made available when the study is complete?

Reviewer #1: Yes

5. Is the manuscript presented in an intelligible fashion and written in standard English?

Reviewer #1: No

6. Review Comments to the Author

You may also provide optional suggestions and comments to authors that they might find helpful in planning their study.

Reviewer #1: Dear authors,

I would like to commend you for revising the manuscript and adding some clarity to the proceedings of research. At the same time, I would urge to the authors to make up following necessary changes in the protocol:

1. Introduction section

i. Please focus this part on the facts according to title of you study. Don’t repeat the WHO guidelines and available treatment options. Explain what the deficiencies or berries in implementations of these guidelines in your area or hospitals are and how you can rectify them.

ii. Line 90-92, please insert the year to which this data belongs, page 3.

iii. Please add reference to the statement made in line 149-150, page 5.

2. Study area section

i. Line 208-215, page 8; This statement is repetition of last paragraph of introduction part. So, it can be omitted.

3. Variable definition sections

i. Please don’t repeat the definition of pre-eclampsia (line 262-264, and line 272-276 at page 10. This section needs to be worked on and rephrased.

ii. Authors still have not made it clear how information will be collected in tool, whether it will be a smartphone or computer based form or a printed questionnaire will be kept at the hospitals.

iii. Similarly, there will be different categories of health care workers like medical graduates, nursing staff and other paramedical staff at each hospital. Authors should explain how they will prevent bias while collecting information from them.

4. Independent variables

i. This section as whole need to be rewritten, sentences is framed very badly. e.g line 298. Page 11; The tool will be adopted from [29] and will be little modified to suit the current study.

Other comments

i. Authors are requested to work on English language in manuscript. Although, they have improved a lot, but much more needs to be done. Please use uniform method of writing pre-eclampsi/eclampsia instead of writing PE/E or full form randomly. Similarly use blood pressure instead of writing Blood Pressure.

ii. Authors have not improved reference section. Some references are incomplete. e.g. Reference number at 4, 6, 7, 9, 18, 21, and 25. Please rewrite them.

Best regards.

7. PLOS authors have the option to publish the peer review history of their article (what does this mean?). If published, this will include your full peer review and any attached files.

Reviewer #1: No

---

## [Author Response · Author response to Decision Letter 1]

5 Jul 2022

1. Introduction section 

i. Please focus this part on the facts according to title of you study. Don’t repeat the WHO guidelines and available treatment options. Explain what the deficiencies or berries in implementations of these guidelines in your area or hospitals are and how you can rectify them. 

Response:This section has been re written line 69-170

ii. Line 90-92, please insert the year to which this data belongs, page 3. 

Response:The year is inserted (line 108-110)

iii. Please add reference to the statement made in line 149-150, page 5. 

Response:The reference is added (line 164)

2. Study area section 

i. Line 208-215, page 8; This statement is repetition of last paragraph of introduction part. So, it can be omitted. 

Response:The paragraph is omitted

3. Variable definition sections 

i. Please don’t repeat the definition of pre-eclampsia (line 262-264, and line 272-276 at page 10. This section needs to be worked on and rephrased. 

Response:The section has been omitted but the important information is included in variables measurement section (line 287-305) 

ii. Authors still have not made it clear how information will be collected in tool, whether it will be a smartphone or computer-based form or a printed questionnaire will be kept at the hospitals 

Response:This information is included in line 346-349

iii. Similarly, there will be different categories of health care workers like medical graduates, nursing staff and other paramedical staff at each hospital. Authors should explain how they will prevent bias while collecting information from them. 

Response:For healthcare providers, self-administered questionnaire will be used. The same standardized questionnaire will be used to all categories and results will be presented for each healthcare category. This information is added in line 316-317.

4. Independent variables 

i. This section as whole need to be rewritten, sentences is framed very badly. e.g line 298. Page 11; The tool will be adopted from [29] and will be little modified to suit the current study. 

Response:The section is re-written line 311-336

Other comments 

i. Authors are requested to work on English language in manuscript. Although, they have improved a lot, but much more needs to be done. 

Response: The manuscript was sent to English editor

Please use uniform method of writing pre-eclampsi/eclampsia instead of writing PE/E or full form randomly. Similarly use blood pressure instead of writing Blood Pressure. 

Response: This has been corrected in the whole document

ii. Authors have not improved reference section. Some references are incomplete. e.g. Reference number at 4, 6, 7, 9, 18, 21, and 25. Please rewrite them. 

Response:References are now corrected (number 4,12,13,15,19,23). After editing the text one reference was dropped

---

## [Decision Letter · Decision Letter 2]

19 Jan 2023

PONE-D-21-13246R2

Auditing the Readiness of Healthcare Facilities in Antenatal Care Referral and Management of Pre-Eclampsia in Zanzibar- A Protocol for Cross Sectional Study

PLOS ONE

Dear Dr. Rashid,

Thank you for submitting your manuscript to PLOS ONE. After careful consideration, we feel that it has merit but does not fully meet PLOS ONE’s publication criteria as it currently stands. Therefore, we invite you to submit a revised version of the manuscript that addresses the points raised during the review process.

We look forward to receiving your revised manuscript.

Kind regards,

Jasbir Singh, M.D.

Guest Editor

PLOS ONE

Additional Editor Comments:

Dear authors,

Thank you for your kind patience and cooperation during entire review process.

Please revise manuscript was per comments received from reviewer/s.

Reviewer 1

I would like to authors for revising the manuscript and adding some clarity to the proceedings of research. At the same time, I would urge to the authors to make up following necessary changes in the protocol:

1. Introduction section

i. Please focus this part on the facts according to title of you study. Don’t repeat the WHO guidelines and available treatment options. Explain what the deficiencies or berries in implementations of these guidelines in your area or hospitals are and how you can rectify them.

ii. Line 90-92, please insert the year to which this data belongs, page 3.

iii. Please add reference to the statement made in line 149-150, page 5.

2. Study area section

i. Line 208-215, page 8; This statement is repetition of last paragraph of introduction part. So, it can be omitted.

3. Variable definition sections

i. Please don’t repeat the definition of pre-eclampsia (line 262-264, and line 272-276 at page 10. This section needs to be worked on and rephrased.

ii. Authors still have not made it clear how information will be collected in tool, whether it will be a smartphone or computer based form or a printed questionnaire will be kept at the hospitals.

iii. Similarly, there will be different categories of health care workers like medical graduates, nursing staff and other paramedical staff at each hospital. Authors should explain how they will prevent bias while collecting information from them.

4. Independent variables

i. This section as whole need to be rewritten, sentences is framed very badly. e.g line 298. Page 11; The tool will be adopted from [29] and will be little modified to suit the current study.

Other comments

i. Authors are requested to work on English language in manuscript. Although, they have improved a lot, but much more needs to be done. Please use uniform method of writing pre-eclampsi/eclampsia instead of writing PE/E or full form randomly. Similarly use blood pressure instead of writing Blood Pressure.

ii. Authors have not improved reference section. Some references are incomplete. e.g. Reference number at 4, 6, 7, 9, 18, 21, and 25. Please rewrite them.

Best regards.

Reviewer 2

PLease see the enclosed file for comments( track changes). TITLE NEEDS REVISION.

SYNTAX / TECHNICAL PROBLEM.

The objective is NOT OK,

Reviewer 3

I must congratulate the authors for planning a very apt study. I have attached my comments as annotation in the attached pdf.

Few of the comments are as follows:

Abstract

• Future tense has to be used at some places.

Introduction

• Minor grammatical corrections required

• The more recent data from Zanzibar should be quoted

Methods

• The bullets could be replaced by plain text

• Sample size estimation

• The various paragraphs could be clubbed

Analysis of the response

• The percentile part has to be removed

• The threshold has to be defined to label as acceptable.

References

• Please correct ref no 4 and 5

the annexes can also be formatted so that they are easier to fill.

Reviewer 4

The study topic is interesting especially in resource poor settings. The study protocol needs language editing and along sentence reframing as mentioned by fellow reviewers.

Please resubmit with revisions.

Reviewer 5

The current study will be a major milestone in determining the prevalence of pre-eclampsia in Zanziber, because the last study done was in the year 2011 (as per the reference given) in just one clinic. Since this study includes many PHCs and referral hospitals, it can give more accurate picture of prevalence of pre-eclampsia as well as the readiness of health facilities in early diagnosis and management. Ensuring dispersal of guidelines for management of pre-eclampsia pregnant women identified within the sample size will be appreciated. That can help in framing better management protocols which will include measures to curtail existing loopholes.

There are grammatical corrections required, cited some:

Abstract (bottom up fifth line) simple grammatical correction: Data will be ‘analyzed’ using SPSS software.

Line 105: may ‘contribute’ to

Line 110: Reframe grammatically if possible

Line 351: the results obtained will portray

Best regards,

Reviewers' comments:

Reviewer's Responses to Questions

**Comments to the Author**

1. Does the manuscript provide a valid rationale for the proposed study, with clearly identified and justified research questions?

Reviewer #2: No

Reviewer #3: Yes

Reviewer #4: Yes

Reviewer #5: Yes

2. Is the protocol technically sound and planned in a manner that will lead to a meaningful outcome and allow testing the stated hypotheses?

Reviewer #2: No

Reviewer #3: Yes

Reviewer #4: Yes

Reviewer #5: Yes

3. Is the methodology feasible and described in sufficient detail to allow the work to be replicable?

Reviewer #2: No

Reviewer #3: Yes

Reviewer #4: Yes

Reviewer #5: Yes

4. Have the authors described where all data underlying the findings will be made available when the study is complete?

Reviewer #2: No

Reviewer #3: No

Reviewer #4: Yes

Reviewer #5: Yes

5. Is the manuscript presented in an intelligible fashion and written in standard English?

Reviewer #2: No

Reviewer #3: Yes

Reviewer #4: Yes

Reviewer #5: Yes

6. Review Comments to the Author

You may also provide optional suggestions and comments to authors that they might find helpful in planning their study.

Reviewer #2: PLease see the enclosed file for comments( track changes). TITLE NEEDS REVISION.

SYNTAX / TECHNICAL PROBLEM.

The objective is NOT OK,

Reviewer #3: I must congratulate the authors for planning a very apt study. I have attached my comments as annotation in the attached pdf.

Few of the comments are as follows:

Abstract

• Future tense has to be used at some places.

Introduction

• Minor grammatical corrections required

• The more recent data from Zanzibar should be quoted

Methods

• The bullets could be replaced by plain text

• Sample size estimation

• The various paragraphs could be clubbed

Analysis of the response

• The percentile part has to be removed

• The threshold has to be defined to label as acceptable.

References

• Please correct ref no 4 and 5

the annexes can also be formatted so that they are easier to fill.

Reviewer #4: The study topic is interesting especially in resource poor settings. The study protocol needs language editing and along sentence reframing as mentioned by fellow reviewers.

Please resubmit with revisions.

Reviewer #5: The current study will be a major milestone in determining the prevalence of pre-eclampsia in Zanziber, because the last study done was in the year 2011 (as per the reference given) in just one clinic. Since this study includes many PHCs and referral hospitals, it can give more accurate picture of prevalence of pre-eclampsia as well as the readiness of health facilities in early diagnosis and management. Ensuring dispersal of guidelines for management of pre-eclampsia pregnant women identified within the sample size will be appreciated. That can help in framing better management protocols which will include measures to curtail existing loopholes.

There are grammatical corrections required, cited some:

Abstract (bottom up fifth line) simple grammatical correction: Data will be ‘analyzed’ using SPSS software.

Line 105: may ‘contribute’ to

Line 110: Reframe grammatically if possible

Line 351: the results obtained will portray

7. PLOS authors have the option to publish the peer review history of their article (what does this mean?). If published, this will include your full peer review and any attached files.

Reviewer #2: **Yes: **AMARJEET SINGH

Reviewer #3: No

Reviewer #4: No

Reviewer #5: No

---

## [Author Response · Author response to Decision Letter 2]

14 Feb 2023

Comments 

Reviewer 1 

1. Introduction section 

i. Please focus this part on the facts according to title of you study. Don’t repeat the WHO guidelines and available treatment options. Explain what the deficiencies or berries in implementations of these guidelines in your area or hospitals are and how you can rectify them. 

Response:This section has been re written line 80-158

ii. Line 90-92, please insert the year to which this data belongs, page 3. 

Response The year is inserted (line 115-118)

iii. Please add reference to the statement made in line 149-150, page 5. Response The reference is added (line 150)

2. Study area section 

i. Line 208-215, page 8; This statement is repetition of last paragraph of introduction part. So, it can be omitted. 

Response The paragraph is omitted

3. Variable definition sections 

i. Please don’t repeat the definition of pre-eclampsia (line 262-264, and line 272-276 at page 10.

 Response This section needs to be worked on and rephrased. The section has been omitted 

ii. Authors still have not made it clear how information will be collected in tool, whether it will be a smartphone or computer-based form or a printed questionnaire will be kept at the hospitals 

Response This information is included in line 338-339

iii. Similarly, there will be different categories of health care workers like medical graduates, nursing staff and other paramedical staff at each hospital. Authors should explain how they will prevent bias while collecting information from them. 

Response For healthcare providers, self-administered questionnaire will be used. The same standardized questionnaire will be used to all categories and results will be presented for each healthcare category. This information is added in line 306-307.

4. Independent variables 

i. This section as whole need to be rewritten, sentences is framed very badly. e.g line 298. Page 11; The tool will be adopted from [29] and will be little modified to suit the current study. 

Response The section is re-written line 300-322

Other comments 

i. Authors are requested to work on English language in manuscript. Although, they have improved a lot, but much more needs to be done. Response The manuscript was sent to English editor

Please use uniform method of writing pre-eclampsi/eclampsia instead of writing PE/E or full form randomly. Similarly use blood pressure instead of writing Blood Pressure. 

Response This has been corrected in the whole document

ii. Authors have not improved reference section. Some references are incomplete. e.g. Reference number at 4, 6, 7, 9, 18, 21, and 25. Please rewrite them. 

Response References are now corrected (number 11,12,14,19,23,24). After editing the text one reference was dropped.

Reviewer 2 

Title needs revision 

Response The title is revised to read: Auditing the Readiness of Healthcare Facilities for Referral and Management of Pre-Eclampsia Cases sin Zanzibar- A study Protocol

Comments on Track changes document: 

Abstract: grammatical errors 

Response Grammar correction was applied to the entire document. 

Introduction: 

• paragraph 2 is a redundant 

Response The paragraph is deleted, line 93-99

• grammatical errors 

Response Grammar correction was applied to the entire document. 

model used: the outcome measure does not link with the process measure 

Response Correction has been made, only two domains of the model will be used, the structure and the process. The outcome is left out, line 168-175

Objectives: 

• need to be corrected 

Response The objective of the prevalence of pre-eclampsia is omitted, line 179-184

Variable measurement

• need revision

• give reference for categorizing the management into proper and improper management 

Response 

• The variable measurement of pre-eclampsia is omitted since the objective was omitted, line 275-279

• We used a score of 100% as proper, less than that is improper, line 289-291.

Data collection: the observation method, how will you control the Hawthorne effect? 

Response: The observation method will be used to assess the availability of equipment and supplies and tests; therefore, Hawthorne effect is not expected, line 367-369

Reviewer 3 

Abstract

• Future tense has to be used at some places. 

Response Future tense is now used, line 49

Introduction 

Minor grammatical corrections required 

Response Grammar correction was applied to the entire document. 

The more recent data from Zanzibar should be quoted 

Response The more available data are now used, line 107-108, 115-116

Methods 

The bullets could be replaced by plain text 

Response The bullets are removed, line 179-184

Sample size estimation 

Response Sample size estimation for pregnant women was removed since the objective was omitted line 228-231

The various paragraphs could be clubbed 

Response Paragraph are reduced in number

Analysis of the response 

The percentile part has to be removed

The threshold has to be defined to label as acceptable. 

Response This statement has been removed, line 305, 314 

References 

Please correct ref no 4 and 5 

Response: Reference has been deleted because the paragraph was deleted. 

Reviewer 4 

The study topic is interesting especially in resource poor settings. The study protocol needs language editing and along sentence reframing as mentioned by fellow reviewers. Please resubmit with revisions. 

Response Grammar correction was applied to the entire document. 

Reviewer 5 

There are grammatical corrections required, cited some:

• Abstract (bottom up fifth line) simple grammatical correction: Data will be ‘analyzed’ using SPSS software.

• Line 105: may ‘contribute’ to

• Line 110: Reframe grammatically if possible

• Line 351: the results obtained will portray 

Response Grammar correction was applied to the entire document. 

• This statement has been corrected, line 70

• The paragraph was deleted, line 127

• The statement has been corrected, line: 132-133

---

## [Decision Letter · Decision Letter 3]

18 May 2023

Auditing the Readiness of Healthcare Facilities for  Referral and Management of Pre-eclampsia Cases in Zanzibar- A Protocol Study

PONE-D-21-13246R3

Dear Dr. Rashid,

We’re pleased to inform you that your manuscript has been judged scientifically suitable for publication and will be formally accepted for publication once it meets all outstanding technical requirements.

Kind regards,

Jasbir Singh, M.D.

Guest Editor

PLOS ONE

Additional Editor Comments (optional):

Reviewers' comments:

Reviewer's Responses to Questions

**Comments to the Author**

1. Does the manuscript provide a valid rationale for the proposed study, with clearly identified and justified research questions?

Reviewer #2: Yes

Reviewer #3: Yes

Reviewer #5: Yes

2. Is the protocol technically sound and planned in a manner that will lead to a meaningful outcome and allow testing the stated hypotheses?

Reviewer #2: Partly

Reviewer #3: Yes

Reviewer #5: Yes

3. Is the methodology feasible and described in sufficient detail to allow the work to be replicable?

Reviewer #2: Yes

Reviewer #3: No

Reviewer #5: Yes

4. Have the authors described where all data underlying the findings will be made available when the study is complete?

Reviewer #2: Yes

Reviewer #3: Yes

Reviewer #5: Yes

5. Is the manuscript presented in an intelligible fashion and written in standard English?

Reviewer #2: Yes

Reviewer #3: Yes

Reviewer #5: Yes

6. Review Comments to the Author

You may also provide optional suggestions and comments to authors that they might find helpful in planning their study.

Reviewer #2: Readiness of Healthcare Facilities – not defined – A COMPOSITE SCORE could have been contemplated?

LINE 77 Introduction T Risk factors like obesity- pl check TYPO / proof reading error.

"Whether management of pre-eclampsia was conducted as per the established guidelines, which includes provision of correct medications, monitoring of blood pressure and fetal heart rate, checking protein in urine and conducting liver and kidney function tests..." –

Will there be an observation of the process? If yes- ETHICAL dilemma!? BIAS…What if NOT ADHERED to? Will it NOT be corrected? ("Process measure will be adherence to the guideline")..

IT CAN NOT BE A descriptive cross-sectional design, e.g., the text states-"When a healthcare facility meets all required referral standards to refer a woman with signs of pre-eclampsia or eclampsia based on the recommended guidelines it is considered a successful referral in this study."

Jhpiego ?, JHPIEGO

Reviewer #3: Thank you for responding to majority of comments.

the threshold of adequate knowledge still needs to be addressed in the Compliance of referral guideline, Knowledge of pre-eclampsia, Skills for management of pre-eclampsia still needs to be addressed.

Reviewer #5: The authors have made necessary changes to their manuscript, remarkably in standard English and grammar. The protocol can be accepted for publication.

7. PLOS authors have the option to publish the peer review history of their article (what does this mean?). If published, this will include your full peer review and any attached files.

Reviewer #2: No

Reviewer #3: No

Reviewer #5: **Yes: **Dr. Madhuri Devaraju

---

## [Editor Report · Acceptance letter]

22 May 2023

PONE-D-21-13246R3 

Auditing the Readiness of Healthcare Facilities for Referral and Management of Pre-Eclampsia Cases in Zanzibar- A Study Protocol 

Dear Dr. Rashid:

I'm pleased to inform you that your manuscript has been deemed suitable for publication in PLOS ONE. Congratulations! Your manuscript is now with our production department. 

Kind regards, 

on behalf of

Dr. Jasbir Singh 

Guest Editor

PLOS ONE